# Impact Analysis of Thermally Pre-Damaged Reinforced Concrete Frames

**DOI:** 10.3390/ma13235349

**Published:** 2020-11-25

**Authors:** Joško Ožbolt, Luka Lacković, Daniela Ruta

**Affiliations:** Institute of construction materials, University of Stuttgart, Pfaffenwaldring 4, 70569 Stuttgart, Germany; luka.lackovic@iwb.uni-stuttgart.de (L.L.); danielaruta@hotmail.com (D.R.)

**Keywords:** reinforced concrete, fire, impact, numerical simulation, thermo-mechanical model, nonlocal temperature, microplane model

## Abstract

In the present study, the influence of thermally induced damage of reinforced concrete (RC) frames on their static and dynamic response is experimentally and numerically investigated. In the experimental test, the RC frame is first pre-damaged through fire exposure and then loaded from the side with the impact of a steel pendulum. To verify the recently developed coupled thermo-mechanical model for concrete, transient 3D FE simulation is carried out. The rate and temperature-dependent microplane model is used as a constitutive law for concrete. It is first shown that the simulation is able to realistically replicate the experimental test. Subsequently, the numerical parametric study is performed where the dynamic and static response of RC frame is simulated for both hot and cold states. It is shown that the pre-damage of RC frame through fire exposure significantly reduces the resistance and changes the response. Finally, it is demonstrated that for the impact load the rate sensitive constitutive law of concrete significantly contributes to the response of RC frame.

## 1. Introduction

Reinforced concrete structures exposed to extreme loading conditions such as fire, impact, blast, earthquake, industrial accidents or their combination is in recent years becoming a more relevant topic of research. Such loading scenarios are characterized with high loading rates, often related with fire outbreak, exposing structures to extremely complex conditions, which can lead to strong and fast degradation of mechanical and physical material properties and failure of structures. Therefore, to make structures more resistant on extreme loading conditions, it is important to investigate the behavior of materials and structures exposed to such extreme loads. In the literature, a large number of studies are available for concrete loaded under high loading rates at ambient temperature and even more on thermally exposed concrete structures. However, there is a relatively low number of studies for the structures that are exposed to combined loading, e.g., impact after fire [1,2,3,4,5,6].

The behavior of concrete that was exposed to high temperature changes significantly. Its mechanical properties, e.g., strength and modulus of elasticity, decrease. At thermal loading there are temperature gradients, which lead to thermally induced stresses. The boundary conditions can prevent deformation due to thermal strains, which cause thermally induced stresses and damage. The load induced thermal strains (thermal creep) and relaxation can also strongly influence the structural behavior [7,8,9,10,11]. Moreover, non-explosive and explosive concrete spalling can occur during fire exposure [12,13,14,15,16,17,18,19], which can significantly decrease the resistance of RC structures.

The influence of loading rate on the behavior of quasi-brittle materials and structures has been intensively studied and better understood in recent years [20,21,22,23,24,25,26,27,28,29,30,31]. It is well known that the resistance, failure mode, crack pattern and crack velocity in concrete are strongly influenced by the loading rate. The rate dependent response of concrete is controlled through three different effects: (i) through the rate dependency of the growing micro-cracks (inertia effects at the micro material level); (ii) through the viscous behavior of the bulk material between the cracks (viscosity due to the water content) and (iii) through the influence of inertia on a meso and macro scale, which comes from different sources [27,28,29,30,31]. In numerical studies (meso- or macro-scale), the first two effects should be accounted for through the constitutive law and, assuming that the resolution of structural discretization is fine enough, the third effect should be automatically accounted for through dynamic analysis. For concrete, which is quasi-brittle material, the first two effects dominate for medium and high loading rates. The influence of inertia dominates in the case of higher loading rates (impact), however, even then the material rate dependency cannot be neglected [31]. Different aspects of inertia need to be distinguished: structural inertia, present even in case of elastic analysis, inertia due to the softening or hardening of material exposed to fast loading rates, and inertia at the crack tip, which is responsible for crack branching phenomena. As a consequence of inertia effects there is typically a progressive increase of resistance (apparent strength) with increase of loading rate. Moreover, the failure mode tends to change from mode-I to shear failure mode and, as shown by Ožbolt et al. [27,28,29,30], the inertia generated as a consequence of damage is the main reason.

When the concrete is before mechanical loading pre-damaged through thermal loading (e.g., fire), there is a significant decrease of mechanical properties, at structural level there is additional damage of concrete due to thermally induced stresses. Recently, it has been demonstrated that through high temperature pre-damaged concrete is less sensitive on the influence of high strain rates [32]. Therefore, the dynamic response of reinforced concrete (RC) structures that are thermally pre-damaged can change significantly when compared to initially undamaged RC structures. Here, the influence of thermally induced damage of RC frame on its dynamic properties is investigated numerically. The frame is first exposed to elevated temperature, cooled down to room temperature and then loaded by the side impact of stiff pendulum. To verify the model, the numerical results are compared with the experimental test [32,33]. It should be noted that these kinds of numerical simulations are extremely complex, computationally demanding, and in the literature there are almost no numerical studies available. The problem is multidisciplinary; one has to consider degradation of mechanical properties of concrete and steel and account for the thermally induced damage due to the free and load induced thermal strains.

The organization of the manuscript is as follows. In the following section, the numerical model, geometry, material properties and loading scenarios are presented. To verify the numerical model, the results of experimental test on one thermally pre-damaged RC frame are compared with the results of the numerical simulation. Subsequently, the results of parametric study are presented and discussed. Finally, the role of rate dependent constitutive law and structural inertia in dynamic (impact) and static analysis of RC frame is investigated and concluding remarks are given.

## 2. Constitutive Law and Finite Element Analysis

### 2.1. Constitutive Law for Concrete

The rate and temperature dependent microplane model is used to model concrete [34,35]. Most of constitutive laws for concrete, formulated in the framework of continuum mechanics, are based on the theories of stress or strain invariants (e.g., plasticity models, damage models or a combination of bots). In contrary to these models, the material response in the microplane model is obtained based on the monitoring stresses and strains in different predefined directions. Each microplane is defined by its unit normal vector *n_i_* (see Figure 1). Microplane strains are assumed to be projections of the macroscopic strain tensor *ε_ij_* (kinematic constraint). On the microplane, considered normal are (*σ_N_*, *ε_N_*) and two shear stress-strain components (*σ_M_*, *σ_K_*, *ε_M_*, *ε_K_*). To realistically model concrete, the normal microplane stress and strain components are decomposed into volumetric and deviatoric parts (*σ_N_* = *σ_V_* + *σ_D_*, *ε_N_* = *ε_V_* + *ε_D_*). Unlike most microplane formulations for concrete, which are based on the kinematic constraint approach, to prevent unrealistic model response for dominant tension (strong localization of strains), kinematic constraint is relaxed [34], i.e., because of strain localization kinematic constraint becomes physically incorrect and has to be relaxed. Based on the micro-macro work conjugacy of volumetric-deviatoric split and using in advance defined microplane stress-strain constitutive laws, the macroscopic stress tensor is calculated as an integral over all predefined microplane orientations:(1)$$\sigma_{ij}=\sigma_{V}\delta_{ij}+\frac{3}{2\pi }\underset{S}{\int }\left[\sigma_{D}\left(n_{i}n_{j}-\frac{\delta_{ij}}{3}\right)+\frac{\sigma_{K}}{2}\left(k_{i}n_{j}+k_{j}n_{i}\right)+\frac{\sigma_{M}}{2}\left(m_{i}n_{j}+m_{j}n_{i}\right)\right]dS$$
where *S* denotes the surface of the unit radius sphere and *δ_ij_* is Kronecker delta.

The microplane model accounts for the two abovementioned strain rate dependent effects: (i) the rate dependency due to the formation (propagation) of micro-cracks that is the effect of inertia at the level of the micro-crack tip, and (ii) the rate dependency coming from the viscous properties of concrete between micro-cracks. For each microplane component, the rate effect is defined according to the rate process theory [35]:(2)$$\begin{array}{cccc} \sigma_{M}\left(\varepsilon_{M}\right)=\sigma_{M}^{0}\left(\varepsilon_{M}\right)\left[1+c_{2}\mathrm{asinh}\left(\frac{\dot{\gamma }}{c_{1}}\right)\right] & \mathrm{with} & \dot{\gamma }=\sqrt{\frac{1}{2}{\dot{\varepsilon }}_{ij}{\dot{\varepsilon }}_{ij}} & c_{1}=\frac{c_{0}}{s_{cr}} \end{array}$$
where *c*_0_ and *c*_2_ are material rate constants, which have to be calibrated by fitting test data, *s_cr_* is assumed spacing of micro-cracks, and ${\dot{\varepsilon }}_{ij}$ components of the macroscopic strain rate tensor. From Equation (2) it can be seen that the rate magnitude is not measured on the individual microplanes, which would not be objective, but on the macro scale and Equation (2) applies to all microplane components except to volumetric compression. This is because for volumetric compression there is no damage, i.e., the material is compacted and therefore rate insensitive. The model parameters from Equation (2) are calibrated based on the uniaxial compressive tests data performed in [25].

The rate sensitivity of concrete (Equation (2)) is assumed to be independent of the thermally induced damage. However, the inertia effects, which have significant influence on dynamic response of concrete, are automatically accounted for through dynamic FE analysis. As mentioned above, after concrete is damaged due to fire exposure the inertia effects become less pronounced [32].

In the here employed thermo-mechanical model for concrete, the total strain tensor consists of three components: mechanical strain, free thermal strain, and load induced thermal strain:(3)$$\varepsilon_{ij}=\varepsilon_{ij}^{m}\left(T,\sigma \right)\;\varepsilon_{ij}^{ft}\left(T\right)+\varepsilon_{ij}^{lits}\left(T,\sigma \right)$$
where *T* is the temperature, $\varepsilon_{ij}^{m}=$ mechanical strain tensor, $\varepsilon_{ij}^{ft}=$ free thermal strain tensor, and $\varepsilon_{ij}^{lits}=$ load induced thermal strain tensor. The mechanical properties of concrete (Young’s modulus, compressive strength, tensile strength and fracture energy) as well as thermal conductivity and heat capacity are temperature dependent [13,16,17]. Usually, in thermo-mechanical models, the material mechanical properties are dependent on the temperature *T* that comes from thermal finite element analysis. However, such approach in quasi-brittle materials such as concrete, which exhibit fracture and damage phenomena, can cause strong localization of thermally induced damage and consequently mesh dependent results. As will be discussed on the numerical example below, this is especially true in dynamic analyses, where the thermally induced damage in the impact zone together with mechanical damage can lead to nonobjective and too strong degradation of structural resistance.

To make the analysis more objective in this respect, an approach in which the thermally dependent mechanical properties of concrete are calculated based on the nonlocal (average) temperature is proposed here. Similarly to the nonlocal strain approach [36,37], the nonlocal temperature $\bar{T}$ is calculated as an average of local temperatures *T* inside representative volume *V_R_*:(4)$$\bar{T}\left(x\right)=\frac{1}{V_{R}\left(x\right)}\underset{V}{\int }\alpha \left(x,s\right)T\left(s\right)dV\left(s\right);\;V_{R}\left(x\right)=\underset{V}{\int }\alpha \left(x,s\right)V\left(s\right)dV\left(s\right)$$
where *α*(*x*,*s*) = nonlocal weighting function, *x* = coordinate of the averaging point, and *s* = coordinate of the contributing point. The adopted nonlocal weighting function is the modified Gauss distribution function (Figure 2):(5)$$\alpha \left(r\right)=\exp\left(-{\left(2.1989\frac{r}{l}\right)}^{2}\right),\;r=\left|s-x\right|;\;l=3d_{a}$$
in which *l* represents the characteristic length, which is assumed to be approximately three times maximum aggregate size (*l* = 3*d_a_*). Note that if the size of the finite elements are significantly larger that the characteristic length or if the temperature field is homogeneous, the nonlocal and local temperatures are the same. Furthermore, free thermal strains and load induced thermal strains are also calculated based on nonlocal temperature, i.e., in Equation (3) *T* is replaced with $\bar{T}$.

### 2.2. Constitutive Law for Steel

Thermo-mechanical model for steel is formulated in the framework of classical rate theory of plasticity. It is based on the von-Mises yield criteria, isotropic hardening and associated flow rule. Its mechanical properties, i.e., Young’s modulus, yield stress and strength, as well as thermal properties (free thermal strains, heat capacity and conductivity) depend on temperature [13,16,17] according to Eurocode 2 [38]. The total strain tensor is decomposed into mechanical strain and free-thermal strain, similar to concrete. Note, however, that after cooling down to the ambient temperature, the mechanical properties of steel are almost fully recovered. Furthermore, temperature dependent properties of steel are calculated using local temperature field obtained from transient heat analysis.

Steel is almost rate insensitive, however, if it is loaded under high loading rates; after threshold strain rate the resistance (apparent strength) starts to increase progressively. The reason is that the non-linear response of steel (strain hardening) that generates strong inertia induced resistance [39]. This effect, as well as other inertia contributions, is automatically accounted for in dynamic finite element analysis.

### 2.3. Finite Element Analysis

The finite element analysis consists of two parts: (i) transient thermo-mechanical and (ii) dynamic analysis. The spatial discretization is performed using standard 3D solid finite elements. Thermal transient analysis is performed using direct integration scheme of implicit type [13,14,15,16,40]. The mechanical part is implicit and based on the Newton–Raphson iteration scheme. The global stiffness matrix is secant, which assures stable convergence for the pre- and post-peak response. In the mechanical part, maximum residual is set to be 1.0% (Euclidian norm). Coupling between the mechanical and non-mechanical part of the model is performed by continuous update of governing parameters during the incremental transient finite element analysis using a staggered solution scheme. After finishing thermo-mechanical analysis, explicit dynamic finite element analysis is conducted. The analysis is performed in the framework of total Lagrange formulation assuming small strains. To obtain analysis objective with respect to the size of finite elements, regularization scheme based on the crack band method is employed [41].

## 3. Experimental Study and Verification of Numerical Model

The experimental tests to investigate the effect of combined thermo-dynamic load on RC frames were carried out at Bhabha Atomic Research Center (BARC) in Mumbai, India [33]. In the tests, the influence of extreme loading conditions with a combination of fire exposure and side impact of pendulum was investigated. To validate the numerical model employed in the subsequent numerical study, only one of these experimental tests is simulated here.

### 3.1. Geometry, FE Discretization and Material Properties

In the experimental test, the realistic size of RC frame was scaled by factor 0.50 and it was designed for gravity load, with no ductile detailing present (see Figure 3). The concrete used in the experiment was a class M25 (mean measured cube strength was approximately 22 MPa) and a steel grade Fe 415, characterized by yield stress of *f_yk_* = 415 MPa. The RC frame structure was fixed on the four contact-point pendulum facility. In the absence of anchorage bars, a supporting structure was made to anchor the frame (see Figure 4).

In numerical finite element simulations, two bodies are involved, the RC frame and the steel impact pendulum. In the first simulation step, the frame is exposed to thermal load, heating and subsequent cooling down to the ambient temperature. The thermo-mechanical analysis is followed by the dynamic analysis (side impact of pendulum). The FE discretization of the complete RC structure is shown in Figure 5a. The steel plate representing the pendulum and the steel reinforcement are shown in detail. The experience with the modeling of concrete-like materials shows that relatively simple constant strain finite elements and randomly generated finite element mesh are most suitable to realistically model fracture and damage phenomena. Moreover, the same elements can also be employed for modeling of reinforcement due to the fact that reinforcement bars are mainly loaded in direct tension or compression. The concrete, main reinforcement, stirrups and pendulum are discretized with standard 4 node constant strain solid finite elements sizing from 8 to 13 mm. For pendulum (steel) and foundation (concrete), linear elastic behavior is assumed. The connection of the pendulum and concrete is performed using contact elements that can transfer only compressive contact forces. Full connection between steel and concrete was assumed (perfect bond). To alleviate the stiffness of the connection between stirrups and main reinforcement, the contact elements (stirrups-main reinforcement) are assumed to have reduced stiffness. The surfaces exposed to elevated temperature are depicted with red color in Figure 5c. Note that also the bottom side of the slab was exposed to fire. In all simulations, one symmetry plane was utilized, as shown in Figure 5d, together with the kinematic constraints of the model. The mechanical and thermal material properties of concrete and steel are summarized in Table 1.

### 3.2. Loading History

As mentioned before, the frame was first exposed to fire, cooled down to ambient temperature, and subsequently laterally loaded with a side impact of pendulum. The fire load was applied following the fire scenario according to the ISO 834 curve (Figure 6) for the duration of 60 min, reaching the maximum ambient temperature of approximately 950 °C (Eurocode 2). Subsequently, the frame was cooled down, with linear decrease of temperature in 120 min, to the ambient temperature of 24 °C with a relatively fast cooling rate of 500 °C/h, where the oven temperature was kept constant for the next 10 h. The first 200 min of the fire loading, compared against the standard ISO 834, is shown in Figure 6.

All of the surfaces except the top slab surface and the four footings, which were thermally insulated, were exposed to fire. The frame was pushed into the oven on a steel trolley (Figure 4). For measuring the temperatures in the oven, 5 thermocouples were placed in the center of the frame and at the center of each bay. Additionally, 80 thermocouples were installed in the frame to measure the temperature distributions across the columns, beams and the slab cross-sections. To measure the temperature along the height of the frame, 5 thermocouples were installed on one of the columns.

In the second loading stage, the thermally damaged frame was laterally loaded with a pendulum under an initial impact angle of 23°. The structure was additionally equipped with strain gauges and accelerometers to measure the strain and acceleration histories at different locations on the frame. The horizontal displacement was measured with high-speed camera, with a frame rate of 1000 fps, which kept track of the stickers placed along the frame and the steel pendulum. The horizontal displacements were measured on the pendulum and at different locations on the frame (see Figure 7a). The displacement measured in the mid of the pendulum (location 4 from Figure 5c) was taken as the displacement loading history applied in the middle of the pendulum in numerical simulation (see Figure 7b). Note that in the numerical analysis only the first 0.15 s of loading history (first rebound) was simulated.

### 3.3. Experimental vs. Numerical Results

In the following, the typical results obtained from the thermo-mechanical and dynamic simulations are shown and compared with the experimental results. Figure 8 shows the distribution of temperature (local) obtained from the transient thermal analysis in column and beam at different locations. Note that heat sources in the furnace cannot on all positions assure the same temperature conditions, i.e., the experiment cannot exactly follow the ISO 834 curve as the analysis does. Bearing this in mind, the agreement between measured and computed temperatures is relatively good, especially for the beam (see Figure 8). The comparison between local and nonlocal temperature distribution is shown in Figure 9. As discussed above, to prevent unrealistic localization of thermally induced damage in concrete, temperature dependent mechanical properties of concrete are assumed to be dependent on nonlocal temperature. In the present study, the characteristic length is taken as *l* = 25 mm. It should be noted that the main differences between local and nonlocal temperature exist close to the exposed surfaces, i.e., in these zones, nonlocal temperatures are slightly lower than the local, due to the relatively large temperature gradients. Local temperature causes unrealistically high thermally induced damage of the concrete surface (impact zone) and, as will be shown below, leads to unrealistic failure of concrete slab in front of the pendulum impact frame surface.

Figure 10 shows the comparison between numerical and experimental results in terms of damage of RC frame after thermal exposure and side impact of pendulum. The red zones represent cracks, in terms of maximum principal strains, which correspond to the crack width equal or greater than *c_w_* = 6 mm. Bearing in mind the complexity of the problem, the agreement between experimental and numerical crack patterns is reasonably good.

Experimentally measured and numerically obtained time displacement histories at different locations are shown in Figure 11. Figure 11a displays the comparison between measured and simulated displacements at the contact between RC frame and pendulum using local and nonlocal temperature, respectively. As can be seen, the agreement between the experiment and simulation for the case when nonlocal temperature distribution was employed in the thermo-mechanical model for concrete is very good. However, if the thermally induced damage of concrete is calculated using local distribution of temperature, the failure takes place too early. As mentioned above, this is due to the unrealistic thermally induced damage of concrete slab, not observed in the experiment. The importance of employing nonlocal temperature when accounting for temperature induced damage of concrete seems to be significant. If the local temperature of the concrete surface is used, which in the present case is in the range of 800 °C, the concrete cover is unrealistically damaged and not able to transfer the impact load as in the experimental test. Therefore, in all further simulations, nonlocal temperature is used to account for more realistic temperature induced damage of concrete. Note, however, that for steel (reinforcement) local temperature has to be employed.

## 4. Numerical Parametric Study

To investigate the influence of thermally induced damage of RC frame under static and dynamic loading on its response and resistance, a parametric numerical study was carried out. The simulations were performed for static and dynamic loading, with and without thermal pre-damage. In the case of dynamic loading, the side impact of pendulum was performed for three different displacement rates. Moreover, the importance of the rate sensitive analysis in static and dynamic analysis was investigated. The geometry, boundary conditions and loading were the same as in the above presented experimental and numerical investigations.

### 4.1. Static Analysis

Static analysis was performed by controlling horizontal displacement at the pendulum mid-line. Considered are the following cases: (i) initially undamaged frame, (ii) thermally damaged frame, hot and cold state after 30 min of fire exposure and (iii) hot and cold state after 60 min of fire duration. The reactions on the frame foundation are shown in terms of reaction-displacement (RD) responses (see Figure 12). The displacement-monitoring node was chosen on the left side of the structure in the contact area of the pendulum and the frame. It is obvious that the thermally induced damage leads to significant decrease in resistance, increase in ductility and decrease of initial stiffness of the frame. The reduction of 22% of the peak reaction between cold and undamaged state can be observed (60 min exposure). However, the response in the hot state exhibits reduction of 42%. The reason for this is the fact that the steel after cooling almost completely restores its strength and stiffness, resulting in higher resistance of the cold state. Furthermore, by analyzing the initial slope of the RD curves, further differences between the hot and cold states can be observed in terms of stiffness. The initial stiffness is mainly governed by damage of concrete. Therefore, the highest reduction is observed for cold state after 60 min of fire exposure. The accumulation of concrete damage after cooling leads to the overall reduction of initial stiffness. Hot conditions are, on the other hand, unfavorable for the steel reinforcement resulting in overall reduction of resistance since the reinforcement has relatively low contribution to the initial stiffness. It is also interesting to observe that for the hot state after 30 and 60 min of fire exposure, the resistance has not significantly changed. This is because the temperature of the main reinforcement in columns critical sections for *t* = 30 and 60 min is not notably different, i.e., 600 °C vs. 730 °C, respectively.

The typical failure modes are shown in Figure 13. The red zones are maximum principal strains, which correspond to the crack width equal or greater than 1.30 mm. It can be seen that in the localization zones (top and bottom of columns) damage is more pronounced for the hot than for the cold state. This is because in hot state, the reinforcement is weaker and consequently, the damage zone is larger and as well as the crack widths.

### 4.2. Dynamic Analysis

In the dynamic analysis, the loading time history was the same as in the experimental test and simulation from Section 3 (see Figure 7b). The initial displacement rate of 1885 mm/s was applied through the control of the mid-line of the pendulum. The same cases are investigated as in above-discussed static analysis. The RC curves are shown in Figure 14 and the predicted peak reactions, together with reactions obtained in static analysis, are summarized in Table 2. Note that displacement is monitored in the same node as in the static analysis (see Figure 12).

Comparing the RD curves, it can be seen that principally the same is valid as for the static analysis, i.e., with increase of temperature induced damage, the peak resistance and initial stiffness decreases, especially for the hot state. The degradation of peak reaction for cold state is lower than in the case of static analysis. When compared to dynamic resistance of undamaged frame with that of thermally pre-damaged (60 min of fire, cold) the degradation is only 16%. Moreover, the reduction of initial stiffness is smaller than in the case of static loading and ductility of the response is not strongly influenced by the thermal damage. However, for the hot state not only concrete but also reinforcement is damaged and consequently, relatively high reduction of dynamic resistance (48%) and initial stiffness is observed for a duration of 60 min of fire. The reason for relatively large differences in the responses of cold and hot state is that in hot state concrete and reinforcement are damaged, however, in cold state the mechanical properties of steel are almost fully recovered.

The comparison between RD curves for static and dynamic loading shows that dynamic resistance is higher and the response stiffer. There are two principal reasons for this: (i) rate sensitive constitutive law for concrete and (ii) inertia effects of different kinds (e.g., inertia due to damage and crack propagation, yielding of reinforcement, structural inertia). In Table 2 the relation between dynamic and static peak resistance of the frame (*RDIF*) is also shown. As can be seen, the dynamic resistance of the frame for different states and given loading conditions is not very different, as it varies between 1.72 and 2.51.

The typical failure modes are shown in Figure 15. The red zones are maximum principal strains, which correspond to the crack width equal or greater than 11 mm. In contrary to the static analysis, damage of the frame is for all loading histories localized in the slab and the top zone of the columns. With increase of thermally induced pre-damage, the size of damage due to impact of pendulum also increases. As shown above, the highest reduction of resistance and initial stiffness was observed for the hot state after 60 min of fire exposure. The reason is thermally induced damage of concrete and reduction of strength and stiffness of reinforcement. The comparison between cold and hot state is shown in Figure 16. Stresses in reinforcement on deformed geometry are also shown. It is obvious that for the hot state deformations, and therefore stiffness, are significantly larger than for the cold state.

### 4.3. Rate Sensitivity vs. Structural Inertia

As discussed above, the dynamic response exhibits higher resistance and stiffness. This is a consequence of the contribution of the rate dependent constitutive law of concrete and inertia effects. To investigate the contribution of these two effects on the dynamic response of the initially undamaged RC frame, additional parametric study is carried out. Static and dynamic analysis with and without rate sensitivity for three different displacement-loading histories of the pendulum (see Figure 17): low (241 mm/s), medium (725 mm/s), and high (1885 mm/s, same as above) are performed. The analyses are carried out only for initially undamaged RC frames.

The typical reaction vs. pendulum-displacement responses are shown in Figure 18. As expected, for quasi-static load the resistance (reaction) only slightly increases with increase of loading rate (Figure 18a). The increase approximately follows the rate dependent constitutive law since the failure mechanism is independent of the loading rate and there are no inertia effects (Figure 19). The concrete strength and fracture energy increases with the increase of loading (strain) rate, according to the rate sensitive constitutive law, and the structural resistance also increases.

However, interesting behavior can be observed for dynamic loading. For relatively high loading rate (1885 mm/s), the rate sensitivity has a significant contribution to the resistance (reaction) and failure mode, i.e., the resistance is approximately double compared to the case without rate sensitivity (Figure 18b). The main reason is due to the different failure modes. Namely, without rate sensitivity the failure takes place in the slab, at the front of the pendulum impact zone (see Figure 20). Therefore, the load is not transferred to the second row of columns of the frame. In contrary to this, when the strain rate is activated, the concrete resistance increases and the transfer of load over the slab to the second row of columns becomes possible. According to the rate dependent constitutive law, tensile and compressive strength become higher than in the case of no rate sensitivity and there is no local failure of concrete slab in the impact zone. Obviously, because of the higher dynamic resistance of the concrete slab, the failure mode changes with the consequence that the dynamic resistance of the frame increases.

For the medium impact velocity (725 mm/s), the impact force can be transferred over the slab in both cases, i.e., with and without rate sensitive constitutive law. Therefore, the effect of rate sensitivity is less important. However, compared to the structural inertia, its contribution is still relatively high. For instance, by comparing RD curves for the loading rate of 725 mm/s (Figure 18c) with the quasi-static RD curve (Figure 18a), it can be concluded that the contribution of strain rate sensitivity is still relatively high. The static resistance is approximately 33 kN, dynamic with rate sensitivity 49 kN, and dynamic without rate sensitivity is 39 kN, respectively. This means that the contribution of rate sensitivity to dynamic resistance is 10 kN and the contribution of inertia 6 kN (33 + 10 + 6 = 49 kN). Of course, this is valid only for this example; however, it principally depends on a number of parameters such as loading rate, structure geometry, size, etc. Figure 18d shows the comparison of RD curves for relatively low loading rate (241 mm/s), which does not lead to the failure of the frame. The peak load is lower than the static resistance and it can be seen that for this case, where the impact load does not cause the collapse of the frame, the effect of rate sensitivity is relatively small.

Comparing the failure modes obtained from quasi-static and dynamic analysis, it can be seen that in static analysis the critical sections are the top and bottom of the columns (Figure 19). However, in the dynamic analysis critical sections are the top zones of the columns and the slab, especially at the front of the impact surface (Figure 20). It is also obvious that the damage is more pronounced if the rate sensitivity is not accounted for (Figure 20).

## 5. Summary and Conclusions

In the present study, static and dynamic 3D FE simulations of initially undamaged and damaged RC frames are carried out. Based on the results the following can be concluded: (i) The comparison between experimental test and numerical simulation shows that the thermo-mechanical transient analysis based on the temperature dependent microplane model is able to realistically replicate the experimental dynamic test of thermally pre-damaged RC frame; (ii) It is shown that instead of local temperature obtained from the standard heat analysis, temperature dependent degradation of concrete mechanical properties should be calculated using nonlocal (average) temperature. This is especially true for dynamic (impact) analysis; (iii) It is demonstrated that fire exposure of RC frames and related thermally induced damage significantly influence their behavior and resistance; (iv) For quasi-static loading, the reduction of structural resistance is the highest for the hot state because the steel reinforcement after cooling almost fully recovers. In the contrary to the peak resistance, the initial stiffness is controlled by concrete properties and it decreases with increasing of fire exposure. Consequently, the initial stiffness is lower for cold state. Ductility of the response is increasing with the increase of fire duration; (v) Compared to quasi-static analysis, in dynamic analysis the reduction of resistance and initial stiffness is principally similar, however, it is significantly higher for the hot state. The overall response is more brittle due to the contribution of inertia; (vi) It is shown that rate sensitive constitutive law for concrete significantly contributes to dynamic structural resistance and failure mode. As expected, with an increase of loading rate, strain rate becomes more important and its influence can be even stronger than the contribution of inertia effects. However, it should be pointed out that its contribution to the overall structural response does not depend only on the loading rate but also on a number of other parameters (e.g., structure type and size, loading, material properties, etc.) and should be investigated for each case separately.

## Figures and Tables

**Figure 1 materials-13-05349-f001:**
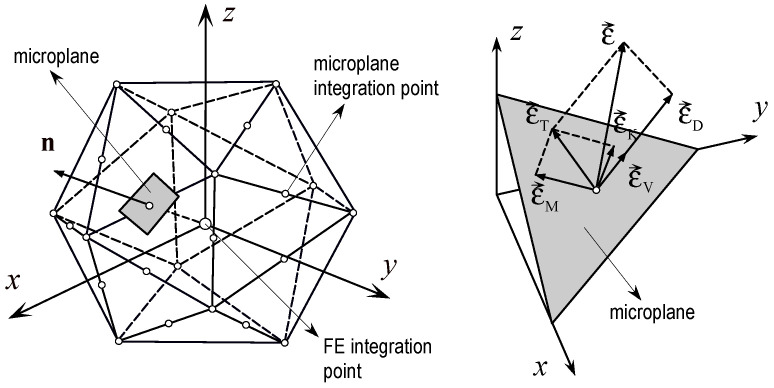
Decomposition of the macroscopic strain vector into microplane strain components—normal (volumetric and deviatoric) and shear.

**Figure 2 materials-13-05349-f002:**
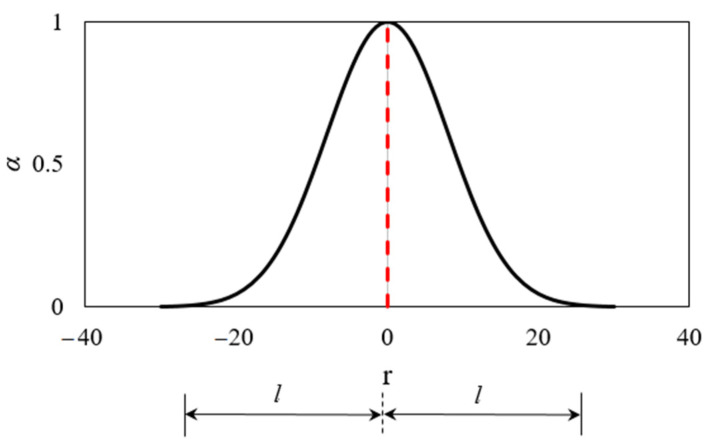
Modified Gauss weight function used for the calculation of nonlocal temperature.

**Figure 3 materials-13-05349-f003:**
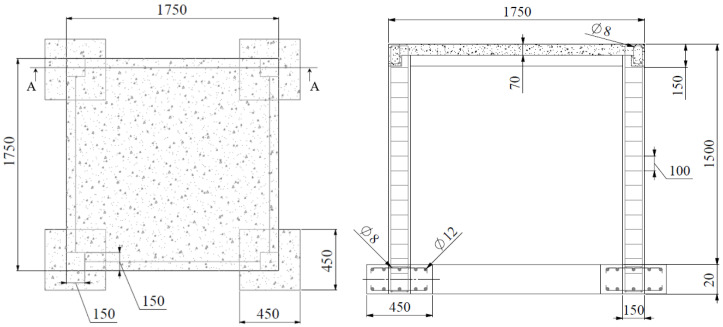
Geometry of the experimentally tested reinforced concrete (RC) frame (Parmar et al. [33]) (all in mm).

**Figure 4 materials-13-05349-f004:**
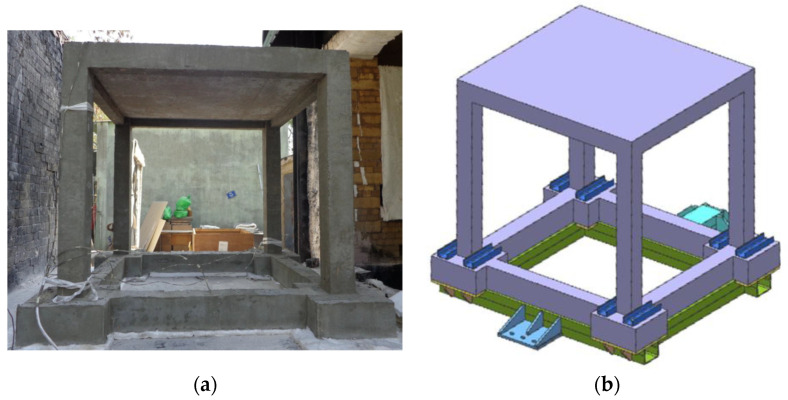
(**a**) RC frame before testing and (**b**) fixing of the base of the frame (Parmar et al. [33]).

**Figure 5 materials-13-05349-f005:**
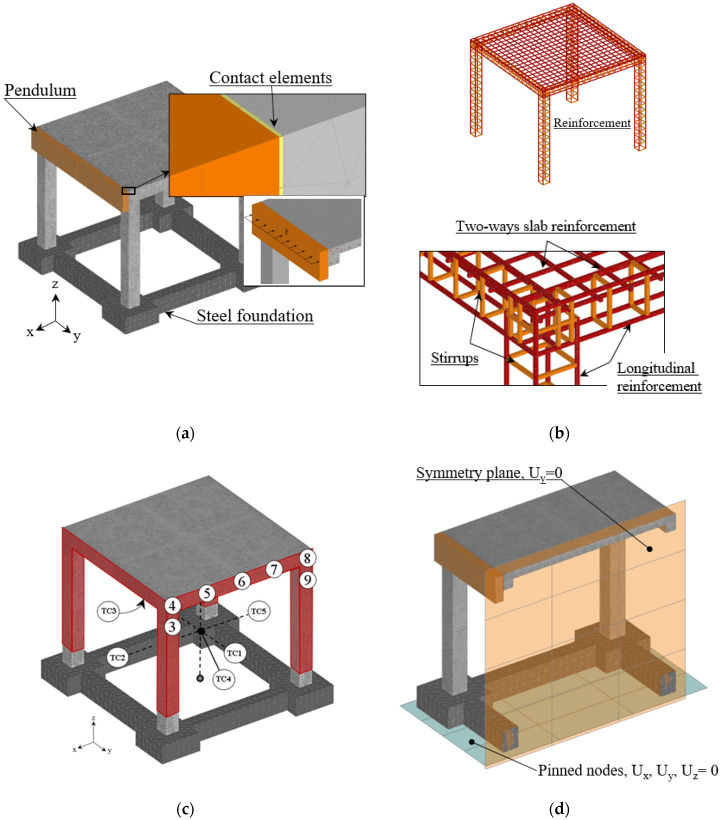
3D finite element discretization of the frame: (**a**) RC frame, pendulum and contact elements, (**b**) main reinforcement and stirrups, (**c**) heated surfaces of the frame in red including bottom surface of the slab and (**d**) symmetry plane and boundary conditions.

**Figure 6 materials-13-05349-f006:**
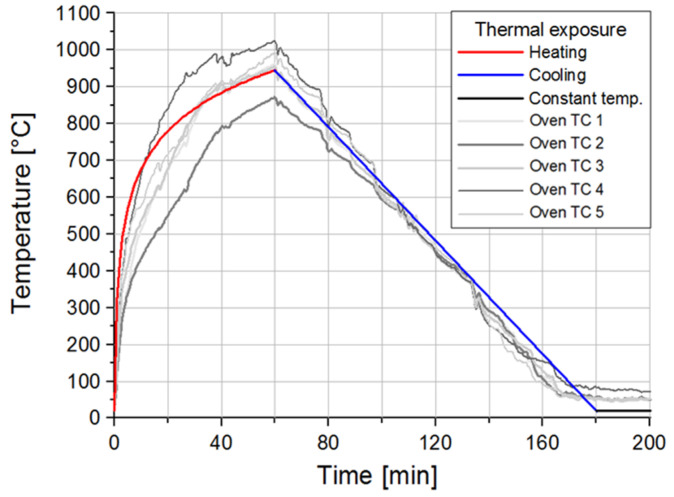
ISO 834 heating curve compared against measured temperature (Parmar et al. [33]).

**Figure 7 materials-13-05349-f007:**
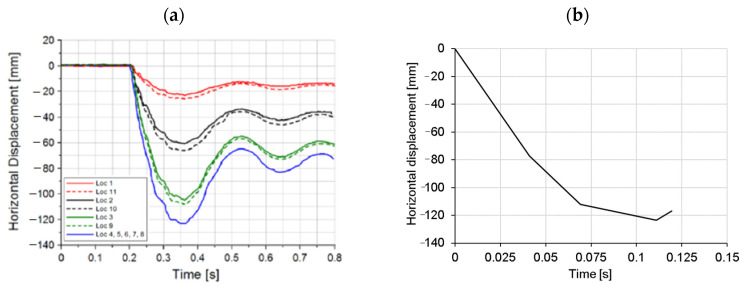
(**a**) In the experiment measured displacement load history (Parmar et al., 2014) and (**b**) displacement load history applied in the finite analysis at the mid line of the pendulum (see Figure 5c).

**Figure 8 materials-13-05349-f008:**
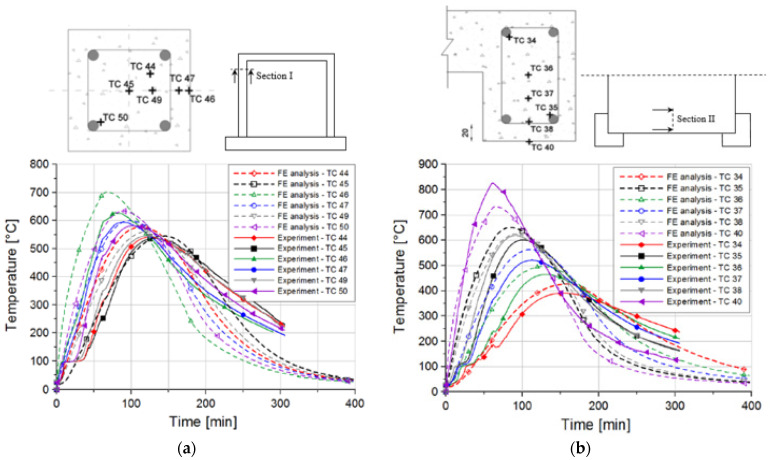
Experimentally measured and numerically obtained distribution of temperatures (local) at different locations: (**a**) column and (**b**) slab.

**Figure 9 materials-13-05349-f009:**
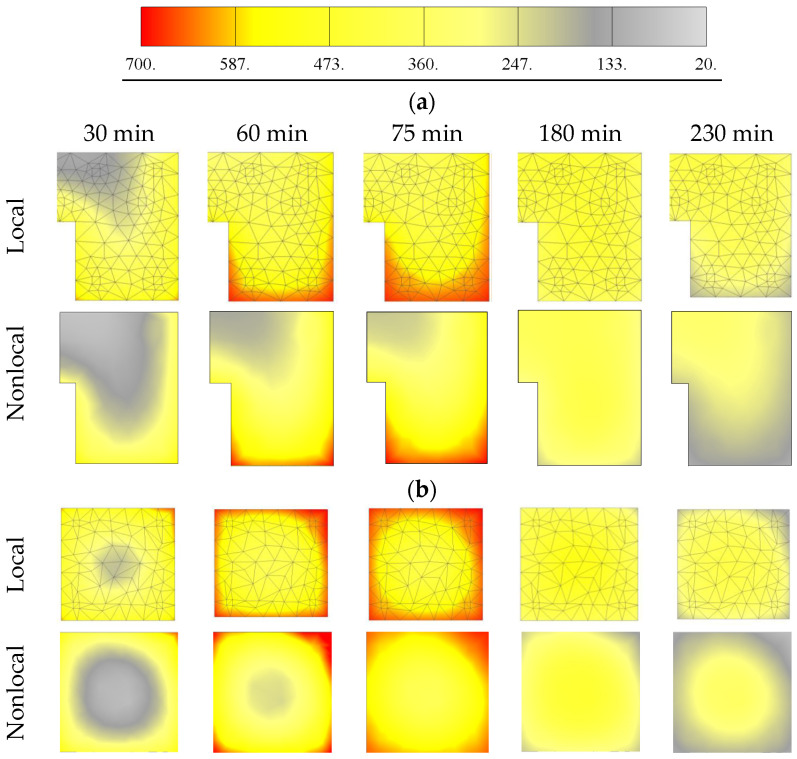
Numerically calculated temperature distributions in: (**a**) beams and (**b**) columns.

**Figure 10 materials-13-05349-f010:**
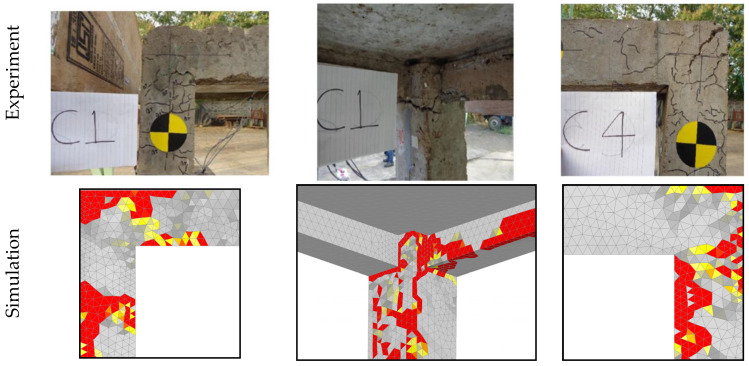
Damage of RC frame after thermal exposure and subsequent side impact of pendulum.

**Figure 11 materials-13-05349-f011:**
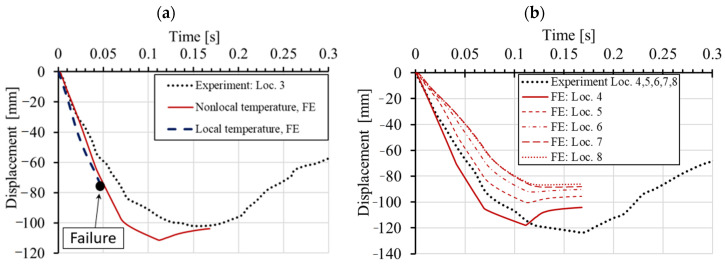
Experimentally measured and calculated displacement time histories: (**a**) Location 3 and (**b**) Locations 4 to 8.

**Figure 12 materials-13-05349-f012:**
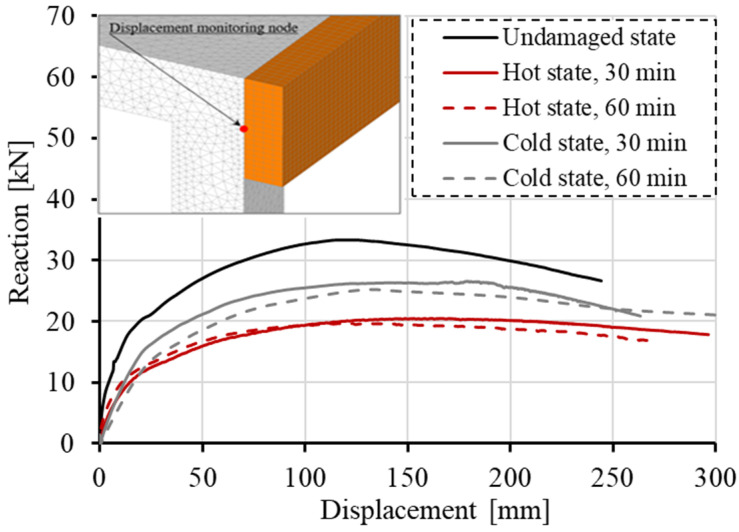
Static analysis—reaction-displacement curves.

**Figure 13 materials-13-05349-f013:**
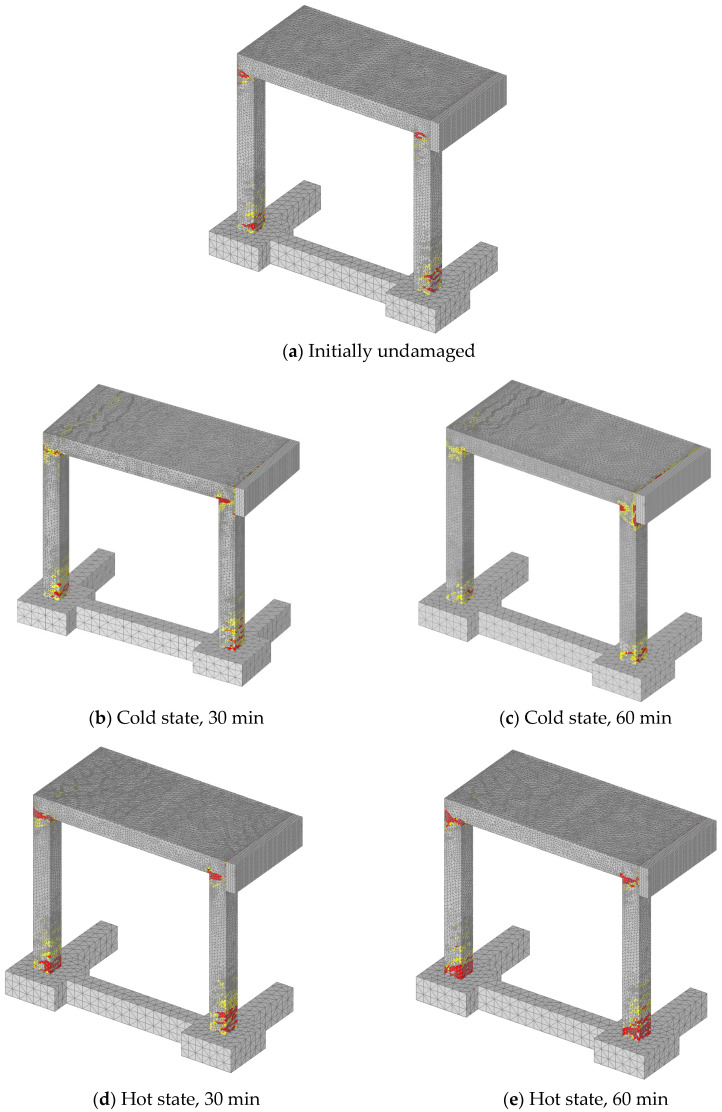
Localization of damage in concrete after peak resistance.

**Figure 14 materials-13-05349-f014:**
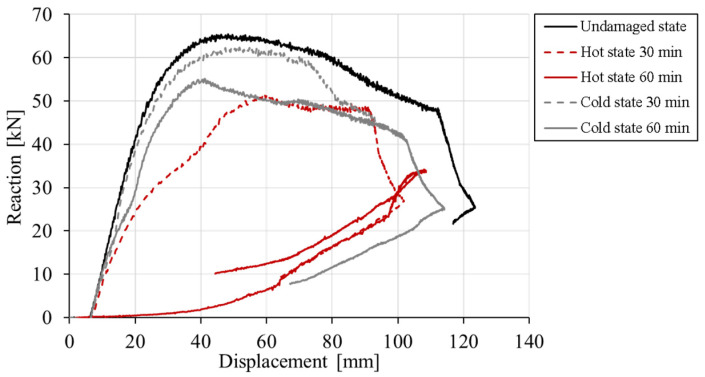
Dynamic analysis—reaction-displacement curves.

**Figure 15 materials-13-05349-f015:**
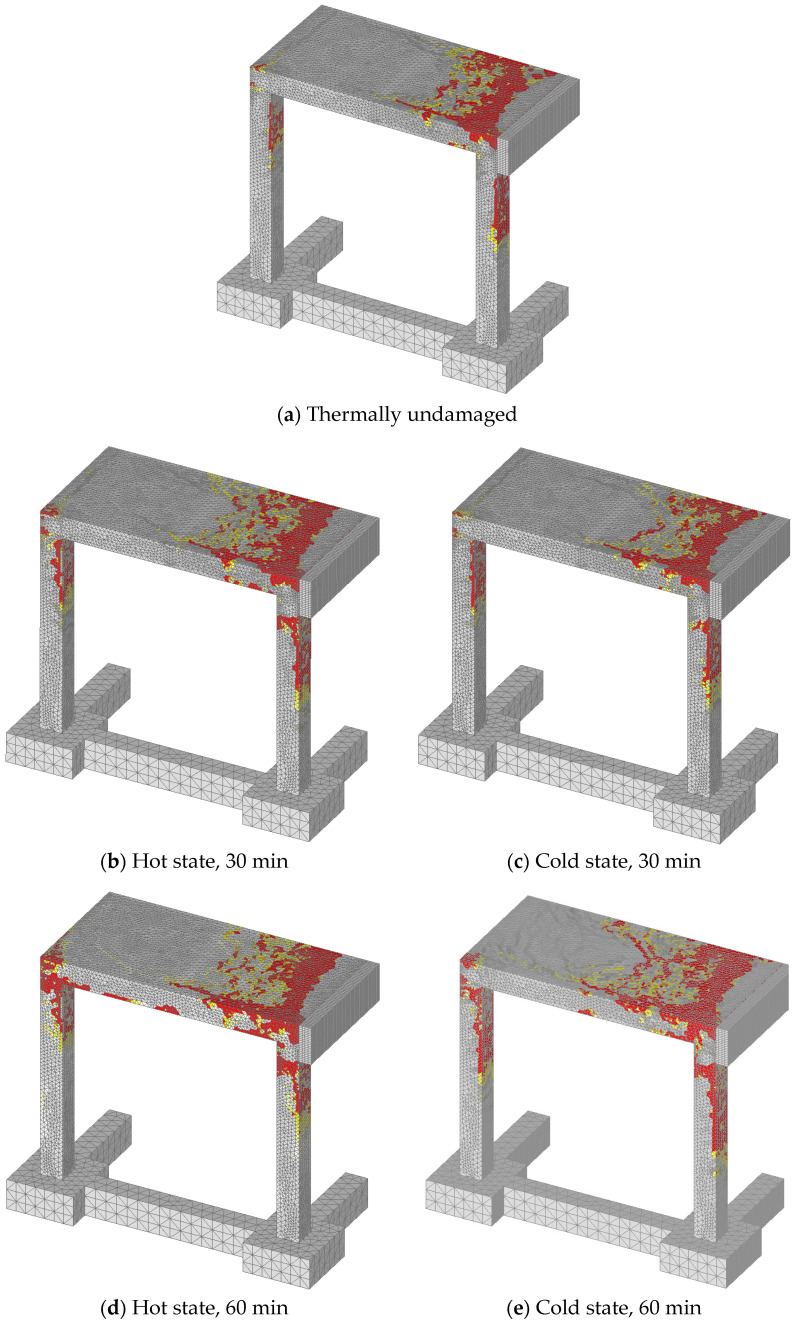
Localization of damage in concrete after peak resistance.

**Figure 16 materials-13-05349-f016:**
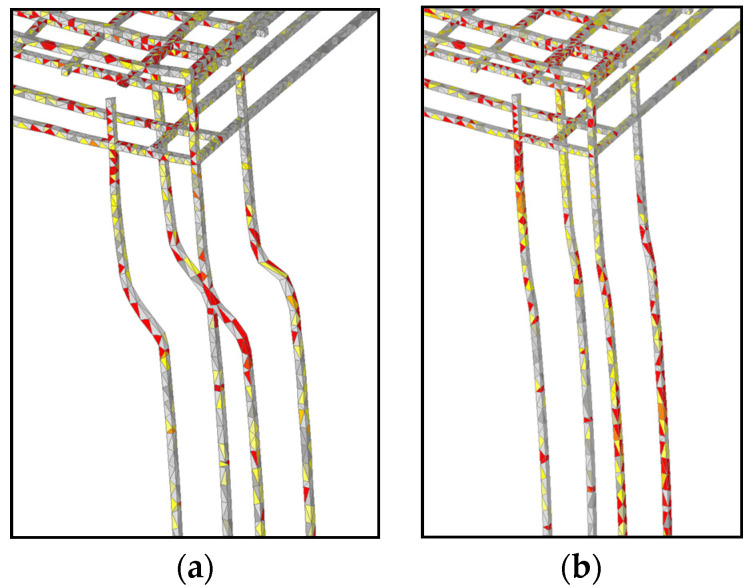
Stresses and deformations in column reinforcement after 60 min of fire exposure at peak reaction: (**a**) hot state and (**b**) cold state; red corresponds to the maximum principal stress of approximately 300 MPa.

**Figure 17 materials-13-05349-f017:**
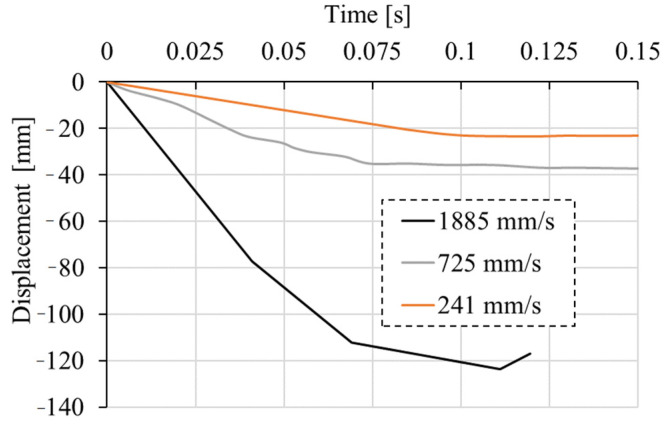
Applied displacement time histories of the pendulum.

**Figure 18 materials-13-05349-f018:**
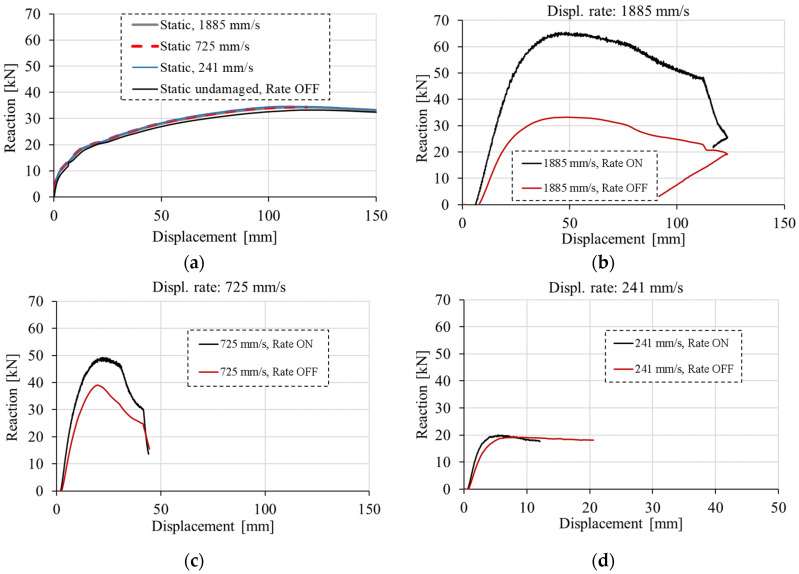
Reaction vs. displacement: (**a**) quasi-static and dynamic for initial loading rates: (**b**) 1885 mm/s, (**c**) 725 mm/s and (**d**) 241 mm/s.

**Figure 19 materials-13-05349-f019:**
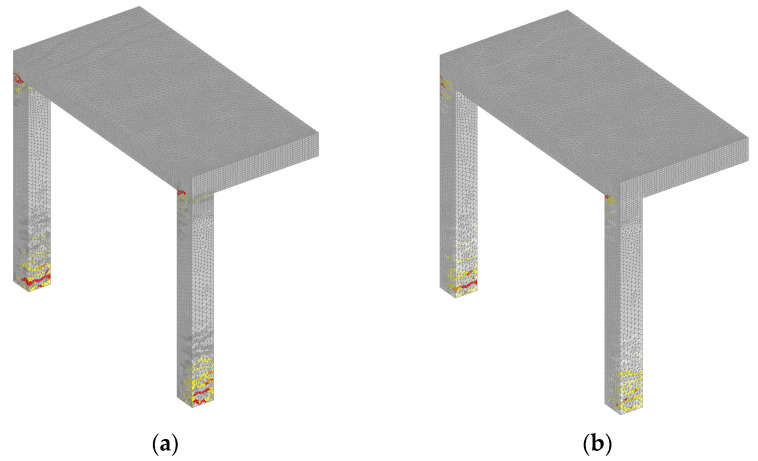
Crack pattern at peak load, quasi-static analysis for loading rate 1885 mm/s: (**a**) no rate sensitivity and (**b**) with rate sensitivity (red = maximum principal strain that corresponds to crack width = 1.30 mm).

**Figure 20 materials-13-05349-f020:**
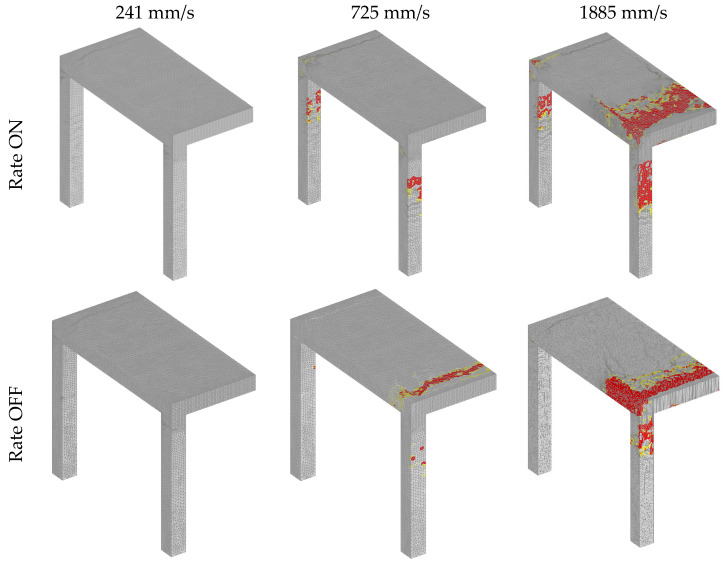
Dynamic analysis, failure modes for different loading rates with and without rate sensitivity (red = maximum principal strain that corresponds to crack width = 3 mm).

**Table 1 materials-13-05349-t001:** Summary of basic material properties for concrete and steel.

Property	Concrete	Reinforcement Steel
Weight density (kg/m^3^)	2400	7800
Young’s modulus (MPa)	26,300	210,000
Compressive (cylinder) strength (MPa)	18.20	-
Tensile strength (MPa)	1.40	-
Fracture energy (J/m^2^)	46.00	-
Poisson’s ratio	0.18	0.33
Yield stress (MPa)	-	480
Strength (MPa)	-	550
Heat capacity (J/kgK)	900	490
Heat conductivity (W/mK)	1.36	43.0

**Table 2 materials-13-05349-t002:** Summary peak reactions for static and dynamic analysis.

State	*R_stat_* (kN)	*R_dyn_* (kN)	*RDIF* (*R_dyn_*/*R_stat_*)
Undamaged	33.28	65.43	1.97
Hot state (30 min)	20.45	51.29	2.51
Cold state (30 min)	26.56	62.46	2.35
Hot state (60 min)	19.70	33.97	1.72
Cold state (60 min)	24.82	55.01	2.22

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
