# Peer review of "Impact Analysis of Thermally Pre-Damaged Reinforced Concrete Frames"

_materials, 2020, doi:10.3390/ma13235349_

Round 1

Reviewer 1 Report

see attached file (comments_auhors.pdf)

Reviewer 2 Report

This paper investigaged the effect of thermal damage of RC on the static and dynamic response of RC frame. Tranient 3D FE simulation was carried out and the results were compared to the gained experimental results.  Although some findings in the manuscript are interesting, I have a couple of major and minor comments that the authors may wish to address:

  1. Line 35, mechanical properties of concrete depends on the heating temperature, with mild heat treatment, concrete’s mechanical properties can be enhanced. Please add some description and refer to the following references.

Suh et al. (2020) Influences of rehydration conditions on the mechanical and atomic structural recovery characteristics of Portland cement paste exposed to elevated temperatures

https://doi.org/10.1016/j.conbuildmat.2019.117453

Masoud Ghandehari; Ali Behnood; and Mostafa Khanzadi Residual Mechanical Properties of High-Strength Concretes after Exposure to Elevated Temperatures

https://doi.org/10.1061/(ASCE)0899-1561(2010)22:1(59)

  1. Line 62, ‘it’ is missing?
  2. Line 69, the info. of the location should be moved to the experimental section.
  3. Line 175, concrete’s strength appeared to be 25MPa. However, the value in Table 1 is quite different (18.2MPa). Please add some description on the differences.
  4. Line 211 to 213, please visualize the locations of installed thermocouples in Fig5.
  5. Line 231, I do not think the numerical results fit very well with experimental results…To me, it looks quite different. In particular, the comparison of T on the surface of concrete (T46). Please clarify it.
  6. Figure 9(a), typos in label (Nonloacl) should be corrected to Nonlocal.
  7. Figure 10, damage of concrete contains cracks due to various factors such as external stress, thermal internal stress (heating), and spalling(heating). The model used in the study can simulate all of the factors?
  8. Figure 13, 15, 19, and 20, is there any experimental results to be compared? I think the stress will focus on the initial damage. Thus, the results shown in Figure 10(initial damage) is critical to static and dynamic simulation results.
  9. Why the shape of steel Rebar in cold state has changed from that in hot state? What is the strain level? Please clarify it.

Round 2

Reviewer 2 Report

The manuscript has been modified based on the reviewers' comments.